# Characteristics of and Public Health Emergency Responses to COVID-19 and H1N1 Outbreaks: A Case-Comparison Study

**DOI:** 10.3390/ijerph17124409

**Published:** 2020-06-19

**Authors:** Qian Wang, Tiantian Zhang, Huanhuan Zhu, Ying Wang, Xin Liu, Ge Bai, Ruiming Dai, Ping Zhou, Li Luo

**Affiliations:** 1School of Public Health, Fudan University, Shanghai 200032, China; wangqian0519@126.com (Q.W.); tiantianzhang18@fudan.edu.cn (T.Z.); 18621600151@163.com (H.Z.); wangying1013@fudan.edu.cn (Y.W.); 19211020044@fudan.edu.cn (X.L.); baige@fudan.edu.cn (G.B.); 17211020050@fudan.edu.cn (R.D.); zhouping@fudan.edu.cn (P.Z.); 2Key Lab of Public Health Safety of the Ministry of Education and Key Lab of Health Technology Assessment of the Ministry of Health, Fudan University, Shanghai 200032, China

**Keywords:** 2019 novel coronavirus, H1N1, emergency response, emerging infectious diseases, public health

## Abstract

Background: Recently, the novel coronavirus disease (COVID-19) has already spread rapidly as a global pandemic, just like the H1N1 swine influenza in 2009. Evidences have indicated that the efficiency of emergency response was considered crucial to curb the spread of the emerging infectious disease. However, studies of COVID-19 on this topic are relatively few. Methods: A qualitative comparative study was conducted to compare the timeline of emergency responses to H1N1 (2009) and COVID-19, by using a set of six key time nodes selected from international literature. Besides, we also explored the spread speed and peak time of COVID-19 and H1N1 swine influenza by comparing the confirmed cases in the same time interval. Results: The government’s entire emergency responses to the epidemic, H1N1 swine influenza (2009) completed in 28 days, and COVID-19 (2019) completed in 46 days. Emergency responses speed for H1N1 was 18 days faster. As for the epidemic spread speed, the peak time of H1N1 came about 4 weeks later than that of COVID-19, and the H1N1 curve in America was flatter than COVID-19 in China within the first four months after the disease emerged. Conclusions: The speed of the emergency responses to H1N1 was faster than COVID-19, which might be an important influential factor for slowing down the arrival of the peak time at the beginning of the epidemic. Although COVID-19 in China is coming to an end, the government should improve the public health emergency system, in order to control the spread of the epidemic and lessen the adverse social effects in possible future outbreaks.

## 1. Introduction

In December 2019, the novel coronavirus disease emerged in Wuhan, a city of 11 million people, and then grew rapidly in China in late January and early February [1,2]. Although the number of infections has since plummeted in China, the disease is now spreading rapidly worldwide, causing heavy loss [3,4]. On 11 March, the World Health Organization (WHO) officially declared the 2019 coronavirus disease (COVID-19) a global pandemic [5], which illustrates the severity of this epidemic. Until 5 April, COVID-19 has infected about 1.2 million people and caused about 66,000 fatalities globally [6].

The last time WHO announced a global pandemic was in 2009, when H1N1 swine influenza broke out worldwide [7]. In March 2009, H1N1 broke out in America and Mexico at almost the same time [8,9], making it difficult to identify the original place and the first emerged case. Although some studies indicated that the origin of H1N1 may be Mexico [10], the first swine influenza A (H1N1) case which has been reported officially and completed the epidemic investigation was from the California in America [9]. After that, the US government’s emergency measures for H1N1 were launched successively. Therefore, in this study, we selected the emergency responses of COVID-19 in China and H1N1 in America as the research objects.

Comparing these two diseases, they share some similarities which may be the reason to cause global pandemic [11,12,13,14,15] (Table 1). First, they are both emerging infectious diseases with no vaccine against them at the time of the outbreak. Second, they can be transmitted by mildly ill or even pre-symptomatic patients, which increases the difficulty of epidemic prevention and control. Third, these two infectious diseases have similar clinical symptoms as common influenza, which makes them difficult to be identified as soon as possible. However, in addition to the particularity of the disease itself, there are many other factors that may affect the outbreak of the global pandemic, such as the speed of the government’s emergency responses and the measures of government’s epidemic prevention and control.

Evidences have already indicated that effective emergency responses of government played an important role in flattening the epidemic curve and slowing down the arrival of peak time, which is beneficial to preventing the diagnosed patients from exceeding the capacity load of the medical system and having no resources to be treated [16]. The efficiency of government’s emergency responses can be reflected by three aspects: (1) hospital reporting stage (time taken for the hospital to report the emerging infectious disease to its national public health agency), (2) pathogen identification and virus gene sequencing stage (time taken for the identification of viral pathogens and the sequencing of virus genes), (3) government policy-making stage (time taken for the establishment of the emergency response department and the implementation of public health policies).

In this study, we mainly focused on the government’s emergency responses to H1N1 and COVID-19. We aimed to compare the key time nodes of government’s emergency responses by using a qualitative retrospective study. We also explored the spread speed and peak time of the two global pandemics by comparing the number of confirmed cases of the two diseases in the same time interval, in order to estimate the effect of government’s emergency responses on flattening the curve and slowing down the peak time of the epidemic.

## 2. Methods

### 2.1. Data Collection

Data for the H1N1 swine influenza in America were collected from the academic literature, announcements from WHO report and data remained on American CDC website [17]. Data for the novel coronavirus in Wuhan were collected from the academic literature, announcements from WHO report, government official websites (National Health Commission of the People’s Republic of China, Chinese Center for Disease control and prevention, Health commission of Hebei Province, Wuhan Municipal Health Commission etc.) [2,18,19,20] and reports from credible media (China central television, YiMagazine, etc.) [21,22].

The confirmed cases of swine influenza in America came from the remained data of American CDC website [23]. The confirmed cases of COVID-19 in China came from DXY.cn [6].

### 2.2. Comparison Content

The timeline of emergency disposal process of COVID-19 was compared with that of H1N1 swine influenza by 6 critical time nodes. They were the hospitalization of the first case, the hospital reporting to the CDC, technical identification of the pathogen, completion of the virus gene sequencing, establishment of the emergency response department and government’s implementation of public health policies.

The speed of three crucial stages of emergency responses, including hospital reporting stage, pathogen identification and virus gene sequencing stage, and government policy-making stage, was also compared to show the efficiency of government’s action on H1N1 and COVID-19. These three stages can be calculated by the 6 critical time nodes with a fixed basis method, using the time point of hospitalization of the first case as the benchmark.

Besides, we also explored the peak time of COVID-19 and H1N1 swine influenza by comparing the confirmed cases in the same time interval (by week).

## 3. Results

After sorting out the government’s emergency responses process of 2009 H1N1 swine influenza and COVID-19, we listed the disposal timeline of important events (Table 2, Figure 1 and Figure 2). We also compared the confirmed cases of COVID-19 and H1N1 swine influenza in the same time interval (Figure 3). The details are as follows.

### 3.1. Hospital Reporting Stage

#### 3.1.1. H1N1 Swine Influenza (2009) 

On 30 March 2009, the first confirmed case (a boy aged 10 years old in San Diego County, CA, USA) of H1N1 went to the outpatient clinic due to fever, cough and vomiting [24]. On 13 April 2009, the American Centers for Disease Control and Prevention (American CDC) was notified of the situation [9].

#### 3.1.2. COVID-19 (2019)

The Wuhan Municipal Health Administration announced that the first confirmed case of novel coronavirus occurred on 8 December 2019 [19]. On 26 December, the Wuhan hospital of traditional Chinese and Western medicine reported 4 cases to the local CDC [22].

The hospital reporting speed was 19 days for COVID-19(2019) and 15 days for H1N1 (2009).

### 3.2. Pathogen Identification and Virus Gene Sequencing Stage

#### 3.2.1. H1N1 Swine Influenza (2009)

When the first case of H1N1 received treatment at the outpatient clinic, a nasopharyngeal swab was collected for testing as part of a clinical study. Initial testing at the clinic identified an influenza A virus, but the test was negative for known human influenza subtypes such as H1N1, H3N2, and H5N1. On 14 April 2009, American CDC received clinical specimens and determined that the virus was swine influenza A (H1N1). On 24 April 2009, American CDC completed the virus gene sequencing of the 2009 H1N1 and uploaded it to a publicly accessible international influenza database, which enabled scientists around the world to use the sequences for public health research.

#### 3.2.2. COVID-19 (2019)

The Wuhan CDC was unable to identify the pathogen on 26 December 2019, and then sent the samples to various testing institutions, including the Shanghai Public Health Clinical Center, the Chinese Academy of Sciences (Wuhan Virus Institute), etc. On 7 January 2020, Chinese CDC isolated a new type of coronavirus from the samples [21]. On 8 January 2020, the National Health Administration confirmed that the virus was novel coronavirus.

On 11 January 2020, the Shanghai Public Health Clinical Center united other testing institutions to complete the virus gene sequencing of COVID-19. On the next day, the national health commission shared the novel coronavirus gene sequences with WHO.

The accumulated days completing pathogen identification and virus gene sequencing after the first confirmed case was 26 days for H1N1 and 35 days for COVID-19 (2019).

### 3.3. Government Policy-Making Stage

#### 3.3.1. H1N1 Swine Influenza (2009)

American CDC activated disease Emergency Operations Center (EOC) on 22 April 2009, to coordinate the response to this emerging public health threat. On 26 April 2009, the United States Government determined that a public health emergency existed nationwide; CDC’s Strategic National Stockpile (SNS) began releasing 25% of the supplies in the stockpile that could be used to protect and treat influenza [17].

#### 3.3.2. COVID-19 (2019)

On 20 January 2020, the command center for epidemic prevention and control was established in Wuhan. On 22 January 2020, the government of Hubei province launched a level-two response to public health emergencies [18].

The government began to implement emergency response policies 46 days after the first case emerged for COVID-19 (2019), and 28 days for H1N1 (Figure 2).

### 3.4. The Epidemic Spread Speed and Peak Time of COVID-19 and H1N1

#### 3.4.1. H1N1 Swine Influenza (2009)

On 30 March 2009, H1N1 swine influenza emerged in America [9]. The cases increased smoothly first and then grew rapidly at about 13–15 weeks (22 June to 6 July) after the first case emerged [25]. About 15 weeks after the disease outbreak, the confirmed cases in America reached the peak, and then decreased steadily (Figure 3).

The second pandemic wave began in the Southeastern United States as children returned to school in mid-August and early September, and peaked in late October. In April 2010, according to the monitoring data of WHO, the influenza epidemic was coming to an end [26]. In total, about 214 countries in the world reported confirmed cases of H1N1 influenza, causing over 18,449 deaths [27]. On August 2010, Margaret Chan, director-general of the WHO, confirmed that the H1N1 influenza pandemic was over.

#### 3.4.2. COVID-19 (2019)

The coronavirus emerged in China on 8 December 2019. About 8–11 weeks (26 January to 16 February) after the disease’s outbreak, the confirmed cases grew rapidly in China and reached the peak (Figure 3). Since then, the number of infections per day has plummeted in China owing in large part to containment efforts [6].

In summary, the first peak time of H1N1 epidemic occurred about 15 weeks after the disease broke out. For COVID-19, it was 11 weeks, which was 4 weeks earlier than that of H1N1. In addition, the H1N1 epidemic curve in America was relatively smoother than COVID-19 in the first 4 months after the disease broke out.

## 4. Discussion

To our knowledge, this was one of the few studies to compare the emergency responses and the spread characteristics of the two global pandemics. In this retrospective comparative analysis, we found that the whole emergency responses process to H1N1 spent about 28 days, which was 18 days faster than that of COVID-19.

In this study, we mainly explored three crucial stages of the emergency responses. The results showed that the most time-consuming stage of COVID-19 was the hospital reporting stage (19 days), which accounted for 41.3% of the whole emergency response process and about 4 days longer than that of H1N1. Besides, the stage of pathogen identification and virus gene sequencing of COVID-19 consumed 5 day longer than H1N1. For the government policy-making stage, COVID-19 used 9 days longer than H1N1. The slow speed of these three stages led to the low efficiency of emergency responses to COVID-19, which could mostly be attributed to two reasons: (1) The initial misjudgment of viral transmission route. The initial epidemiological investigation did not find the evidence for human-to-human transmission of COVID-19, which misled the public and caused the government to miss the best time to take proper measures to prevent the spread of the disease. This mistake might be attributed to the insufficient number and expertise of public health professionals in China [28]. (2) The lack of effective report system of emerging infectious disease. In China, a surveillance information system of infectious diseases was established to monitor the existing 39 infectious diseases to ensure real-time reporting of emerging cases. However, for a new virus not included in the report catalogue, the report system was unable to perform its due functions, which indicated the lack of timely report mechanism of emerging infectious disease in China.

In contrast, the faster response of HIN1 was due to several objective and subjective reasons. (1) Human infection with swine influenza had occurred before. During 2005 to 2009, 12 cases of human infection with swine influenza were reported in America [29,30,31], which brought research basis for this disease and might be beneficial to accelerating the pathogen identification and virus gene sequencing speed. (2) Simultaneous outbreaks in multiple states. In March and April 2009, H1N1 emerged in more than one place almost at the same time, which attracted more focus from the government on preventing the infectious disease outbreak. (3) Close collaboration between clinical laboratories and public health laboratories in America can detect emerging infections timely. During H1N1, clinical laboratories forwarded untypable pathogens to the American CDC laboratories timely. The laboratories confirmed the emergence of a new strain of influenza, which triggered a massive public health response.

We also compared the confirmed cases of H1N1 and COVID-19 in the same interval. The results showed that the peak time of H1N1 came about 4 weeks later than COVID-19. The epidemic curve of H1N1 was flatter than COVID-19 after the disease emerged. These results illustrated that the timely emergency responses to H1N1 flattened the curve and slowed down the arrival of peak time at the beginning of the epidemic, which left more time for America to control the spread of disease.

However, affected by the seasonal characters of H1N1, the second wave came quickly after the first peak time came to an end. In contrast, during COVID-19, China adopted several strict measures to contain the epidemic, for example, voluntary plus mandatory quarantine, stopping mass gatherings, closure of educational institutes or places of work where infection has been identified, and isolation of households, towns, or cities [32]. What happened in China later showed that these measures effectively contained the rapid spread of epidemic. However, the potential economic impact of those stringent measures could be substantial, which could have been avoided through timely and effective epidemic emergency responses. Although COVID-19 in China is coming to an end, the lessons came from China should be learned by other countries. At the same time, China should take measures continually to be alert to the resurgence of the epidemic.

This study has several potential limitations. First, we used six time nodes to explore the process of the government’s emergency responses, which might be limited when evaluating the situation of a major pandemic. Then, the data mostly came from official reports, and the number of confirmed cases was far below the actual value due to a large number of patients with mild illness did not receive hospitalization, especially for the H1N1 epidemic. However, the overall transmission trend of the disease is not affected, so the results of this study are credible.

## 5. Conclusions

The speed of the emergency responses to H1N1 was faster than COVID-19, which might be an important influential factor for flattening the epidemic curve and slowing down the arrival of the peak time at the beginning of the epidemic. Although COVID-19 in China is coming to an end, the government should improve the public health emergency system, in order to control the spread of the epidemic and lessen the adverse social effects in the possible future outbreaks.

## Figures and Tables

**Figure 1 ijerph-17-04409-f001:**
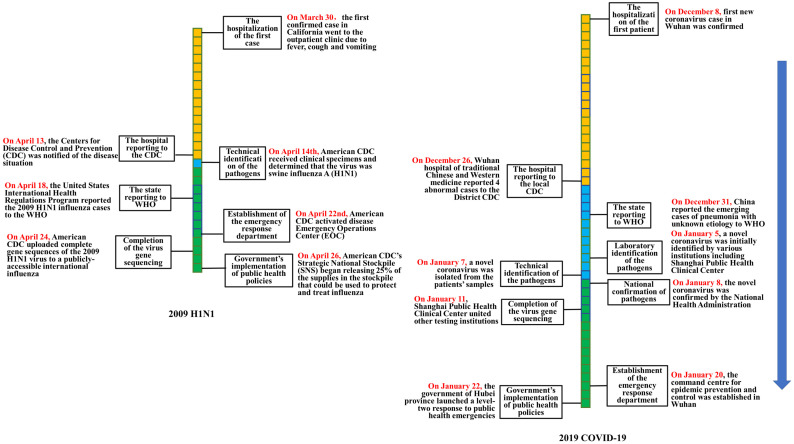
Comparison of the emergency disposal timeline of H1N1 swine influenza (2009) and COVID-19 (2019).

**Figure 2 ijerph-17-04409-f002:**
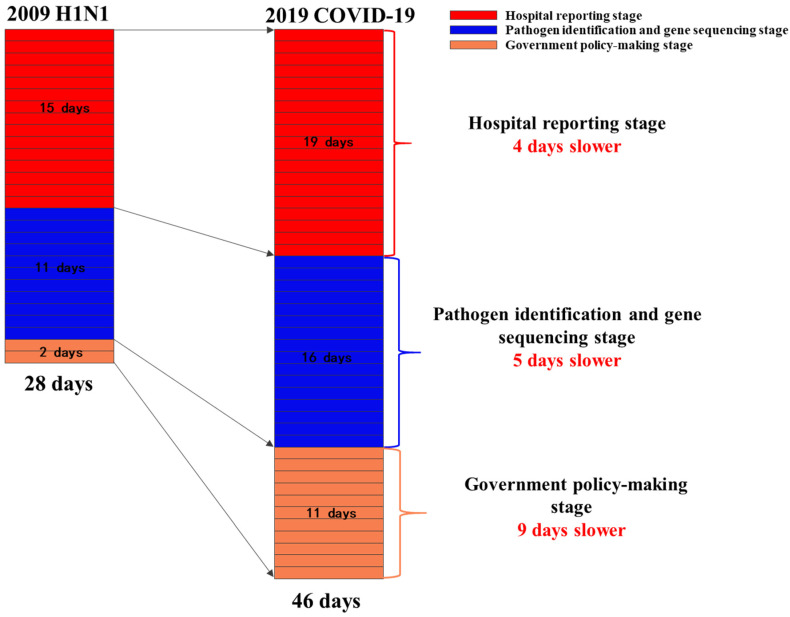
Comparison of three critical speeds between H1N1 swine influenza (2009) and COVID-19.

**Figure 3 ijerph-17-04409-f003:**
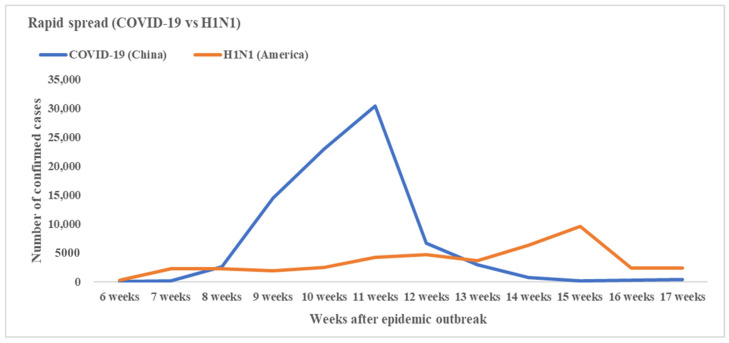
Number of total confirmed cases of COVID-19 in China and H1N1 in America the same time interval after the first emerged cases (by week).

**Table 1 ijerph-17-04409-t001:** Characteristics of the H1N1 swine influenza and COVID-19.

Characteristics	H1N1	COVID-19
Susceptible population	People younger than 30	People aged 30–79 years old
Main route of transmission	Droplets or fomites	Droplets or fomites, contact
Common clinical symptoms	Fever, cough, sore throat and myalgia	Fever, cough, short of breath
Seasonality	Yes	Unknown
Diagnosis	RT-PCR	RT-PCR
Human-to-human transmission	Yes	Yes
Vaccine	Lack	Lack

**Table 2 ijerph-17-04409-t002:** Emergency disposal timeline of H1N1 swine influenza (2009) and novel coronavirus (2019).

Timeline	Three Crucial Stages	H1N1 Swine Influenza (2009)		COVID-19 (2019)	
		Dates and Events	Accumulated Days	Dates and Events	Accumulated Days
**The hospitalization of the first case**	Hospital reporting stage	On 30 March, the first confirmed case in California went to the outpatient clinic due to fever, cough and vomiting.	1	On 8 December, the first novel coronavirus case in Wuhan was confirmed.	1
**The hospital reporting to the CDC**		After receiving the patients, the clinic reported the cases to the San Diego Health Bureau, but the specific time was not reported. On 13 April, American CDC was notified of the disease situation.	15	On 26 December, the Wuhan hospital of traditional Chinese and Western medicine reported four abnormal cases to the District CDC.	19
**Technical identification of the pathogens**	Pathogen identification and virus gene sequencing stage	On 14 April, American CDC received clinical specimens and determined that the virus was swine influenza A (H1N1).	16	On 7 January, Chinese CDC isolated a new type of coronavirus from samples collected from patients.	31
**Completion of the virus gene sequencing**		On 24 April, American CDC uploaded the complete gene sequences of the 2009 H1N1 virus to a publicly accessible international influenza database.	26	On 11 January, Shanghai Public Health Clinical Center united other testing institutions to decipher the virus genome.	35
**Establishment of the emergency response department**	Government policy-making stage	On 22 April, American CDC activated the disease Emergency Operations Center (EOC).	24	On 20 January 2020, the command center for epidemic prevention and control was established in Wuhan.	44
**Government’s implementation of public health policies**		On 26 April, American CDC’s Strategic National Stockpile (SNS) began releasing 25% of the supplies in the stockpile that could be used to protect and treat influenza.	28	On 22 January, the government launched a level-two response to public health emergencies.	46

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
