# Peer review of "Characteristics of and Public Health Emergency Responses to COVID-19 and H1N1 Outbreaks: A Case-Comparison Study"

_ijerph, 2020, doi:10.3390/ijerph17124409_

Round 1
Reviewer 1 Report
*The manuscript is both interesting and very relevant to the readers using an innovative approach.
*I suggest minor language editing
*I may suggest that the authors discuss why they think that H1N1 response was faster, i.e. explain the reasons.
Very well done!
Author Response
Dear reviewer,
Thank you for your encouragement and valuable suggestions!
According to your comments we have revised the manuscript (ijerph-796280), in the following we listed the responses to your comments point by point. By the way, changes were highlighted by using the “Track Changes” function in Microsoft Word in the revised manuscript.
Comment 1: Minor language editing
Response: Thank you for your kind reminding! We are very sorry for the language problem and the inconvenience it caused to your reading. Each of our co-authors has read the manuscript again and checked the language carefully. We have edited the manuscript and corrected some grammatical errors and typing mistakes. We hope it can meet the standard of publication this time. Besides, we will check the final script again or use a professional English editing service if necessary.
Comment 2: discuss why H1N1 response was faster, i.e. explain the reasons.
Response: Thanks for your instructive suggestions! I totally agree with you and add the reasons to discuss why H1N1 response was faster. Besides, we also explained the possible reasons why the COVID-19 response was slower. Please see Line 202-223 in simple markup version.
Besides, we have proofread and improved the other parts of the manuscript. The main revised parts were colored by yellow in the revised manuscript. We hope it can meet the standard of publication this time.
Thanks again for your helpful suggestions and spending time on the review.
Best regards,
Authors
Reviewer 2 Report
The authors described the timeframe for reporting of H1N1 and COVID-19. The authors compared the response of two governments to two different outbreaks.
Perhaps, the rapid identification of H1N1 in 2009 was due to prior experiences of prior infections as the authors indicated.
The response to the two pandemics was very different and a lockdown did not happen during the H1N1 pandemic. Until now, it is unclear how effective lockdown is in controlling COVID-19.
Authors should elaborate more why there was a delay in diagnosis and reporting of COVID-19
Author Response
Dear reviewer,
Thank you for your helpful comments and suggestions!
According to your comments we have revised the manuscript (ijerph-796280), in the following we listed the responses to your comments point by point. By the way, changes were highlighted by using the “Track Changes” function in Microsoft Word in the revised manuscript.
Comment: Authors should elaborate more why there was a delay in diagnosis and reporting of COVID-19
Response: Thank you for your valuable suggestions. We have revised this part and explained the possible reasons why the COVID-19 response was slower. Besides, we have also stated the possible reasons why H1N1 response was faster. Please see Line 202-223 in simple markup version.
Besides, we have proofread and improved the other parts of the manuscript. The main revised parts were colored by yellow in the revised manuscript. We hope it can meet the standard of publication this time.
Thanks again for spending time on the review.
Best regards,
Authors
Reviewer 3 Report
I liked the motivation of the authors behind conducting this study. The central message ("The emergency responses to H1N1 was faster than that of COVID-19") of this work, as per my understanding, may stand out somehow to potential readers, the authors do not seem to see it that way. Please look at the Conclusions section. It does not convey this key message clearly. In addition, the last sentence ("While the strict follow-up measures for COVID-19 in China help to constain the further spread of epidemic effectively in a short time.") is an incomplete sentence and does not help the key message to come out clearly. Besides, this last sentence has typing mistakes, which would have been avoided, had the authors done a good proofreading.
Similarly, the 2nd sentence of the Background section of Abstract is not really a part of background. It says what the authors did do, which means it should be a part of the Methods section.
The usage of acronym "CDC" is confusing throughout. In the case of the 2009 H1N1 pandemic, it refers to the Centers for Disease Control and Prevention in the USA (for which the acronym "CDC" is supposedly to be used, I think). In the case of COVID-19 pandemic, the acronym CDC refers to the Chinese CDC. I encourage the authors to take care of this.
What do the authors mean by "data remained on government official websites" in line 85? In my view-point, the authors should provide proper references to the data sources that were accessed by them for this study.
The subsection title "Pathogen Inspection and public speed" under the Results section sounds a little clumsy to me. Should it not be "Pathogen's genome sequencing and its availability in public domain" or something along this line?
The titles of 6 nodal events, given in Table 2 (under the header Timeline), seems to differ greatly from the corresponding titles under the Results section. The authors are encouraged to make the titles matching smooth.
In addition, the order of the appearance of COVID-19 and 2009 H1N1 in the Results section (here COVID-19 is followed by 2009 H1N1) was reversed in Table 2, Figure 1 and Figure 2. In fact, Table 1 also has the 2009 H1N1 first, which is followed by COVID-19. Also, note Table 1 has a typing mistake. In Table 1, what do the authors mean by older children? Can they give the age-group?
In the second paragraph of the Discussion section, the authors seem to have been trying to justify the delay of 31 days because the world did not have a prior experience with the COVID-19 pathogen (as we had with 12 cases of human infection with swine influenza). True. But, I think, we should be ready for this kind of outbreak situation. Remember, the SARS outbreak in 2002 did not have given us a chance to have any prior experience. Also, I wonder, why did the authors not include any mention of the government response timelines for the 2002 SARS outbreak.
The last but not the least, as I went through the whole text, I realize that there are typos sprinkled throughout the manuscript, including the bibliography section. Proofread, please.
Author Response
Dear reviewer,
Thank you for your helpful comments and suggestions!
According to your comments we have revised the manuscript (ijerph-796280), in the following we listed the responses to your comments point by point. By the way, changes were highlighted by using the “Track Changes” function in Microsoft Word in the revised manuscript.
Comment 1: The central message ("The emergency responses to H1N1 was faster than that of COVID-19") of this work, as per my understanding, may stand out somehow to potential readers, the authors do not seem to see it that way. Please look at the Conclusions section. It does not convey this key message clearly. In addition, the last sentence ("While the strict follow-up measures for COVID-19 in China help to constain the further spread of epidemic effectively in a short time.") is an incomplete sentence and does not help the key message to come out clearly. Besides, this last sentence has typing mistakes, which would have been avoided, had the authors done a good proofreading.
Response: Thank you for your advice! The main purpose of our manuscript is to compare the emergency responses of COVID-19 and H1N1 and find the possible learning points we can take away from these two tragedies. However, the previous Conclusions section didn’t emphasize the main focus. We have rewritten the Conclusions section in the revised manuscript. please see Line 247-251 in simple markup version.
Besides, we are sorry for the language problem and the inconvenience it caused to your reading. Each of our co-authors has read the manuscript again and checked the language carefully. The grammatical errors and typing mistakes found in the article have been corrected in the revised version.
Comment 2: Similarly, the 2nd sentence of the Background section of Abstract is not really a part of background. It says what the authors did do, which means it should be a part of the Methods section.
Response: You are right!Thanks for your kind reminding. We have revised the 2nd sentence of the Background section and explained the reason why we did this research. Please see Line 12-15 in simple markup version.
Comment 3: The usage of acronym "CDC" is confusing throughout. In the case of the 2009 H1N1 pandemic, it refers to the Centers for Disease Control and Prevention in the USA (for which the acronym "CDC" is supposedly to be used, I think). In the case of COVID-19 pandemic, the acronym CDC refers to the Chinese CDC. I encourage the authors to take care of this.
Response: Thank you for your advice. It is really a good idea. In our revised manuscript, we used “American CDC” and “Chinese CDC” separately as the acronyms in order to distinguish the Centers for Disease Control and Prevention in different country.
Comment 4: What do the authors mean by "data remained on government official websites" in line 85? In my view-point, the authors should provide proper references to the data sources that were accessed by them for this study.
Response: Thank you for your kind reminding. We have provided references to the data sources in the revised mauscript, please see line 87 and 94.
Comment 5: The subsection title "Pathogen Inspection and public speed" under the Results section sounds a little clumsy to me. Should it not be "Pathogen's genome sequencing and its availability in public domain" or something along this line? The titles of 6 nodal events, given in Table 2 (under the header Timeline), seems to differ greatly from the corresponding titles under the Results section. The authors are encouraged to make the titles matching smooth.
Response: Thanks for your careful reading and valuable advice. We have checked and rewritten the relevant part carefully in order to better convey the main idea of our manuscript. The efficiency of government’s emergency responses was studied by three main stages: hospital reporting stage, pathogen identification and virus gene sequencing stage, and government policy-making stage. Please see line 63-69. Besides, we have also proofread the article to avoid the matching problems. Please see Line 63-69, 97-108, 114, 126, 148, and Table 2.
Comment 6: In addition, the order of the appearance of COVID-19 and 2009 H1N1 in the Results section (here COVID-19 is followed by 2009 H1N1) was reversed in Table 2, Figure 1 and Figure 2. In fact, Table 1 also has the 2009 H1N1 first, which is followed by COVID-19.
Response: Thanks for your reminding. We have reordered the results section to match the order in other parts of our manuscript. Please see Line 109-183, Table 1 (line 81), Table 2 (line 185), Figure 1 (line 186)and Figure 2 (line 188).
Comment 7: Also, note Table 1 has a typing mistake. In Table 1, what do the authors mean by older children? Can they give the age-group?
Response: Sorry for the language problem. We have checked the languge and revised this part. Besides, we have given the age-group of susceptible population. Please see Table 1 (line 81).
Comment 8: In the second paragraph of the Discussion section, the authors seem to have been trying to justify the delay of 31 days because the world did not have a prior experience with the COVID-19 pathogen (as we had with 12 cases of human infection with swine influenza). True. But, I think, we should be ready for this kind of outbreak situation. Remember, the SARS outbreak in 2002 did not have given us a chance to have any prior experience. Also, I wonder, why did the authors not include any mention of the government response timelines for the 2002 SARS outbreak.
Response: Thank you for your valuable suggestions. You are right, we only stated some objective reasons to explain why the H1N1 was faster without exploring the problems reflected by the delay of emergency responses of COVID-19. It is not comprehensive. We have revised this part and explained the possible reasons why the COVID-19 response was slower. Besides, we have also stated the possible reasons to discuss why H1N1 response was faster. Please see Line 202-223.
As for the timelines for SARS, actually, we listed the timelines of SARS before, but later we found that an article has already compared the timelines of SARS and COVID-19 (Callaway E, Cyranoski D, Mallapaty S, Stoye E, Tollefson J. The coronavirus pandemic in five powerful charts. Nature. 2020;579(7800):482‐483. doi:10.1038/d41586-020-00758-2). Therefore, we deleted this part to avoid some repetition problems. Besides, the epidemics of COVID-19 and H1N1 showed many similarities, they were announced to be global pandemic, and the time interval of them was not too long. Therefore, we selected these two epidemics as the main study objects in our study, in order to find several learning points can be taken away from these two pandemics in the event of future outbreaks.
Comment 9: The last but not the least, as I went through the whole text, I realize that there are typos sprinkled throughout the manuscript, including the bibliography section. Proofread, please.
Response: We felt greatly sorry for the language errors. In the revised version, each of our co-authors has proofread the manuscript again and correted some grammatical errors and typing mistakes. We hope it can meet the standard of publication this time. To ensure the quality of the article, we will check the final script again before publication or use a professional English editing service if necessary.
Thanks again for spending time on the review. Your valuable suggestions did help a lot for us to improve our manuscript.
Best regards,
Authors
Round 2
Reviewer 2 Report
I have no more comments.
Author Response
Dear reviewer,
Thanks a lot for spending time on the review.
Because of your valuable suggestions, we have proofread and improved the manuscript better.
Best regards,
Authors
Reviewer 3 Report
First of all, I thank the authors for their time and work for revising the previous manuscript. The revised manuscript looks great. There are still a few things that, I think, need revision.
The first-time use of the acronym "CDC" in line 65 seems to be inappropriate. First of all, what the authors refer to by CDC is not clear because it is not defined prior (i.e., in the previous 64 lines) to its use. Second of all, even if I ignore that many potential readers can associate or assume it to be referring to the USA's Centers for Disease Control and Prevention, here its use in the context it has been used seems to be country-agnostic (meaning, it should be independent of the USA). For this reason, and given that not every country in the world has named its national public health agency whose acronym can be safely assumed to be CDC (for example, the USA's CDC equivalent in India is called the Public Health Foundation of India), I would recommend that the use of CDC should be replaced with a phrase "its national public health agency" or something along this line.
The web links (for example, for Refs [17, 18]) in the main text should be given the bibliographic section associated the corresponding references.
The use of CfDCaP (and any acronym like this) in the bibliographic section needs to be corrected. It is unnecessary there. Delete it. If I am not mistaken, the acronym CfDCaP is created by the authors by taking the first letter of each word of this phrase: "Centers for Disease Control and Prevention." The acronym CDC itself is used internationally (and therefore sufficient) for the Centers for Disease Control and Prevention.
What is CCTV? Is it closed-circuit television?
Author Response
Dear reviewer,
Thank you for spending time on reviewing our manuscript.
According to your comments we have revised the manuscript (ijerph-796280), in the following we listed the responses to your comments point by point. By the way, changes were highlighted by using the “Track Changes” function in Microsoft Word in the revised manuscript.
Comment 1: The first-time use of the acronym "CDC" in line 65 seems to be inappropriate. First of all, what the authors refer to by CDC is not clear because it is not defined prior (i.e., in the previous 64 lines) to its use. Second of all, even if I ignore that many potential readers can associate or assume it to be referring to the USA's Centers for Disease Control and Prevention, here its use in the context it has been used seems to be country-agnostic (meaning, it should be independent of the USA). For this reason, and given that not every country in the world has named its national public health agency whose acronym can be safely assumed to be CDC (for example, the USA's CDC equivalent in India is called the Public Health Foundation of India), I would recommend that the use of CDC should be replaced with a phrase "its national public health agency" or something along this line.
Response: Thank you for your comment. We didn’t consider the fact that not all the national public health agencies can be called Centers for Disease Control and Prevention. Therefore, the use of the acronym "CDC" may lead to confusion to some readers. Thanks for your reminding, we have revised it in our manuscript, please see line 66. Besides, we also checked the other acronyms in our manuscript to confirm that each of them had been defined prior to its use.
Comment 2: The web links (for example, for Refs [17, 18]) in the main text should be given the bibliographic section associated the corresponding references.
Response: Thank you for your suggestion. We have given the corresponding references for this part. Please see line 86-94.
Comment 3: The use of CfDCaP (and any acronym like this) in the bibliographic section needs to be corrected. It is unnecessary there. Delete it. If I am not mistaken, the acronym CfDCaP is created by the authors by taking the first letter of each word of this phrase: "Centers for Disease Control and Prevention." The acronym CDC itself is used internationally (and therefore sufficient) for the Centers for Disease Control and Prevention.
Response: Thank you for your suggestion. You are right. The acronym CfDCaP was created automatically by the endnote sofware. We have deleted it manually in the revised manuscript.
Comment 4: What is CCTV? Is it closed-circuit television?
Response: Sorry for the confusing acronym. CCTV is the acronym of the national television in China. We have replaced “CCTV” to its full name “China central television”, please see line 92.
Thank you again for your valuable suggestions.
Best regards
Authors